# Measurements and simulations of microtubule growth imply strong longitudinal interactions and reveal a role for GDP on the elongating end

Joseph M Cleary[1], Tae Kim[2], Annan SI Cook[1], Lauren A McCormick[2], William O Hancock[1]*, Luke M Rice[2]*

[1]Department of Biomedical Engineering, Pennsylvania State University, State College, United States; [2]Departments of Biophysics and Biochemistry, The University of Texas Southwestern Medical Center, Dallas, United States

*For correspondence:
woh1@psu.edu (WOH);
Luke.Rice@UTSouthwestern.edu
(LMR)

**Competing interest:** The authors declare that no competing interests exist.

**Abstract** Microtubule polymerization dynamics result from the biochemical interactions of αβ-tubulin with the polymer end, but a quantitative understanding has been challenging to establish. We used interference reflection microscopy to make improved measurements of microtubule growth rates and growth fluctuations in the presence and absence of GTP hydrolysis. In the absence of GTP hydrolysis, microtubules grew steadily with very low fluctuations. These data were best described by a computational model implementing slow assembly kinetics, such that the rate of microtubule elongation is primarily limited by the rate of αβ-tubulin associations. With GTPase present, microtubules displayed substantially larger growth fluctuations than expected based on the no GTPase measurements. Our modeling showed that these larger fluctuations occurred because exposure of GDP-tubulin on the microtubule end transiently 'poisoned' growth, yielding a wider range of growth rates compared to GTP only conditions. Our experiments and modeling point to slow association kinetics (strong longitudinal interactions), such that drugs and regulatory proteins that alter microtubule dynamics could do so by modulating either the association or dissociation rate of tubulin from the microtubule tip. By causing slower growth, exposure of GDP-tubulin at the growing microtubule end may be an important early event determining catastrophe.

## Editor's evaluation

Using improved optical imaging of growing microtubules and improved data analysis, the authors find that microtubule growth fluctuations are less pronounced than previously reported and that GTP hydrolysis in microtubules increases these fluctuations. A mathematical model of microtubule growth suggests that occasionally exposed GDP at the microtubule end can be responsible for increased growth fluctuations. This work provides new mechanistic insight into the molecular mechanism of microtubule growth.

## Introduction

The microtubule cytoskeleton is essential for the proper organization of the interior of eukaryotic cells: microtubules build the mitotic spindle that mediates faithful chromosome segregation during cell division (*Prosser and Pelletier, 2017*), they mediate organelle positioning and help establish cellular polarity, and they form tracks for motor-based transport (*Desai and Mitchison, 1997*). Microtubules are dynamic, multi-stranded polymers that grow and shrink by addition and loss of αβ-tubulin subunits

from the ends of the hollow, cylindrical polymer (*Cleary and Hancock, 2021*; *Desai and Mitchison, 1997*; *Gardner et al., 2013*; *Howard and Hyman, 2003*). The dynamic properties of microtubules are central to their function, and result from the structural and biochemical properties of individual αβ-tubulin subunits and how they interact with the microtubule lattice. It has been challenging to define the quantitative mechanisms of microtubule dynamics, because the microtubule end is a complex biochemical environment where individual tubulins adopt different conformations and can contact variable numbers of neighboring subunits (*Brouhard and Rice, 2018*; *Gudimchuk and McIntosh, 2021*). Consequently, there is yet to be a consensus about the fundamental mechanisms governing microtubule growing, shrinking, and switching. This lack of understanding in turn limits our ability to understand how microtubule regulatory factors and/or microtubule-targeting drugs modulate polymerization dynamics.

An established approach for studying microtubule dynamics is to fit biochemical or mechanochemical models to measurements of microtubule growing and shrinking rates and frequencies of catastrophe and rescue. Early analyses of microtubule growth performed linear fits to measured concentration-dependent microtubule growth rates to estimate the apparent association and dissociation rate constants for tubulin:microtubule binding (*Bergen and Borisy, 1980*; *Mitchison and Kirschner, 1984*; *Walker et al., 1988*) reviewed in *Cleary and Hancock, 2021*. This '1D' model provides a simple, phenomenological way to describe changes in microtubule dynamics caused by microtubule-associated proteins (MAPs) (*Brouhard et al., 2008*; *Chen et al., 2019*; *Geyer et al., 2018*; *McAlear and Bechstedt, 2022*; *Strothman et al., 2019*). However, the 1D model assumes that all tubulin interactions at the microtubule end are equal, whereas it is well established that the microtubule end presents multiple binding sites that differ in affinity (*Atherton et al., 2017*; *Chrétien et al., 1995*; *McIntosh et al., 2018*). 2D 'lattice' models explicitly represent the microtubule protofilaments and thereby can provide a more realistic biochemical model for the microtubule end (*Chen and Hill, 1983*; *Margolin et al., 2012*; *Piedra et al., 2016*; *VanBuren et al., 2002*). Using a small number (~4–5) of fitting parameters, these models can recapitulate measured growing and shrinking rates, and catastrophes emerge naturally as a consequence of GTPase activity. These biochemical models do not capture all observed aspects of microtubule dynamics (e.g. the concentration dependence of catastrophe, and the ability of depolymerizing microtubules to do mechanical work), but they provide a relatively simple minimal framework for the molecular mechanisms that underlie microtubule dynamics.

In principle, fluctuations in microtubule growth rate contain complementary information to constrain models. Early attempts to measure the fluctuations in microtubule growth used an optical trap assay in which the elongating end of the microtubule was grown against a barrier so that movements of the microtubule reflected growth or shrinkage at the end (*Kerssemakers et al., 2006*, *Schek et al., 2007*). These studies revealed that growth episodes were punctuated by small phases of rapid shrinking that had not previously been observable by light microscopy. Microtubule growth fluctuations were subsequently studied by total internal reflection fluorescence (TIRF) microscopy in a study that argued for a model where tubulin interactions at the tip occur rapidly (*Gardner et al., 2011*).

In prior work, we measured interactions between individual αβ-tubulin and the microtubule end (*Mickolajczyk et al., 2019b*) using a combination of nanogold-labeled yeast αβ-tubulin and interferometric scattering (iSCAT) microscopy (*Ortega-Arroyo and Kukura, 2012*; *Young and Kukura, 2019*). Those measurements revealed that the association and dissociation of individual αβ-tubulins occurred more slowly than had been expected based on an earlier analysis of microtubule growth rates and fluctuations (*Gardner et al., 2011*). However, the extent to which these apparently contradictory findings were in conflict was difficult to assess: in addition to using different methods to determine growth fluctuations, the two studies used different sources of αβ-tubulin (fungal for the single-molecule measurements, mammalian brain for the study of fluctuations).

In the present study, we used interference reflection microscopy (IRM) (*Mahamdeh and Howard, 2019*; *Mahamdeh et al., 2018*) to observe the in vitro growth rates and fluctuations of bovine brain microtubules at high temporal resolution. We performed measurements in the presence of GTP (where the microtubules can undergo catastrophe as a consequence of GTPase activity) and in the presence of the hydrolysis-resistant GTP analog GMPCPP (to provide biochemically simpler measurements for analysis). Growth in GMPCPP showed a very small critical concentration, reflecting high affinity interactions between αβ-tubulin subunits and the microtubule end, as well as small concentration-dependent

fluctuations in growth. These characteristics were recapitulated in a biochemical model featuring slow association and dissociation kinetics and in which relatively strong head-to-tail interactions between αβ-tubulins dictate more ragged and tapered microtubule end configurations. Meanwhile, growth in GTP resulted in substantially larger concentration-dependent fluctuations in growth rate than expected from extrapolation of the GMPCPP measurements. These larger magnitude fluctuations could not be recapitulated by the 'GTP only' model of microtubule elongation that provided a good description of the GMPCPP measurements. However, when it allowed for GTP hydrolysis, the model produced fluctuations in growth rates comparable to those observed experimentally. These larger, GTPase-induced fluctuations in simulations resulted from transient, growth inhibitory effects of GDP-tubulin at the end of one or more protofilaments. Collectively, these results support a model of microtubule growth in which tubulin binding dynamics are relatively slow and dominated by longitudinal interactions, and in which exposure of terminal GDP-tubulin regulates the addition of incoming tubulin subunits at the plus-end.

## Results

### When GTPase activity is blocked, microtubules show very low fluctuations in growth rate

We used IRM (*Mahamdeh and Howard, 2019*; *Mahamdeh et al., 2018*) to measure the dynamics of bovine brain microtubules with high temporal resolution (10 frames/s) and good signal-to-noise ratio (exceeding 4). To provide templates for microtubule elongation, our assay used biotinylated, double-cycled GMPCPP 'seeds' attached to a cover slip using neutravidin (*Chen et al., 2019*; *Figure 1A*; inset shows a representative image of a microtubule). To measure the position of microtubule ends over time with sub-pixel resolution, we adapted a previously described algorithm that fits the decay in image intensity at the microtubule tip (*Demchouk et al., 2011*; *Gardner et al., 2011*; *Prahl et al., 2014*; *Figure 1—figure supplement 1*).

Our first series of measurements used the hydrolysis-resistant GTP analog GMPCPP to measure the concentration dependence of microtubule growth rates and fluctuations in a biochemically simple setting, without complications associated with GTPase activity. Representative kymographs and 'tracks' of microtubule plus-end length as a function of time at the low (0.5 µM) and high (1.5 µM) ends of the tubulin concentration range are shown in *Figure 1B*; the high limit was chosen to avoid conditions where microtubules formed spontaneously. We measured growth rates from a large number of tracks at each tubulin concentration (*Figure 1C* shows the resulting average growth rates), obtaining a range comparable to that observed in a prior study (*Gardner et al., 2011*). The measured growth rates followed the expected linear relationship between mean growth rate and tubulin concentration, yielding an apparent on-rate constant for microtubule binding ($k_{on}^{app}$, slope) of 3.1 $\mu M^{-1}s^{-1}$ and apparent critical concentration (x-intercept) of ~50 nM.

The label-free IRM imaging provided long recordings and high temporal resolution. These rich datasets allowed us to analyze fluctuations in microtubule growth rates at the level of individual microtubules, rather than having to concatenate shorter recordings from multiple microtubules, as was done in previous work (*Gardner et al., 2011*). To ensure that the fluctuations reported were solely from microtubule growth, we used fiduciary marks on the slide to remove contributions from sub-pixel stage drift (*Figure 1—figure supplement 1*; *Leduc et al., 2007*). Analyzing microtubule growth fluctuations of individual growth episodes provided better insight into the range of growth properties compared to the analysis of concatenated growth episodes (*Figure 1—figure supplement 2*). Because separating the growth rate from growth fluctuations can be difficult when growth lengths are relatively short, as for our GMPCPP data, we evaluated two methods for quantifying fluctuations around the mean growth rate (*Figure 1—figure supplement 3*). One method obtains growth rates and fluctuations concurrently by fitting mean squared displacement (MSD) vs. time curves (*Gardner et al., 2011*), while the other determines growth rates and fluctuations independently by analyzing the mean and variance of incremental length change distributions, sampled at different time intervals (*Castle et al., 2019*; *Figure 1D*). Separate determination of growth rate and variance provided the most robust results (*Figure 1—figure supplement 3*).

Our analysis reveals that microtubule growth fluctuations in GMPCPP were very small, ranging from 2 to 4 nm²/s (*Figure 1E*). These values are ~5-fold smaller than those measured in a prior study

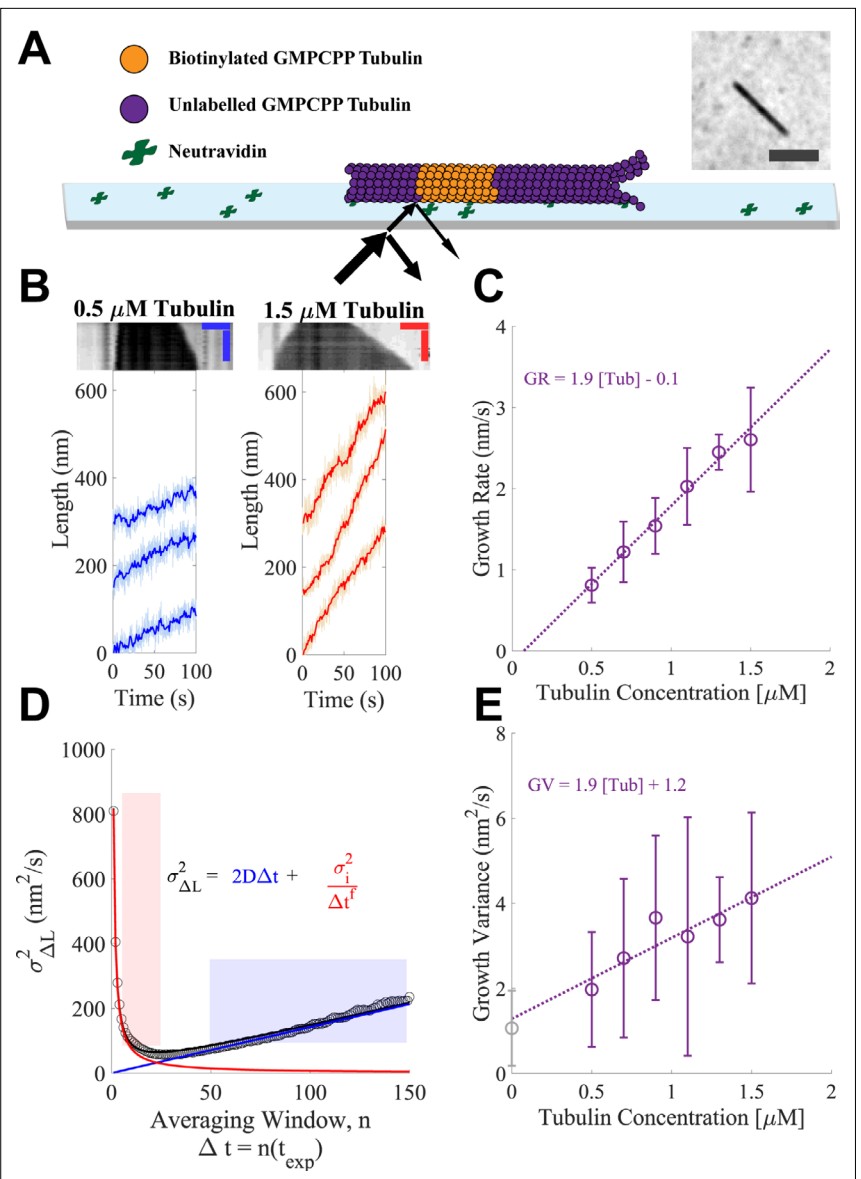

**Figure 1.** Microtubules growing in GMPCPP show slow growth and small fluctuations. (**A**) Schematic of the dynamics assay. Biotinylated bovine GMPCPP microtubule seeds (orange) are attached to the cover slip through neutravidin, free unlabeled bovine tubulin is added to the flow cell in the presence of 1 mM Mg-GMPCPP, and growing microtubules are visualized using interference reflection microscopy (IRM) (image in inset; scale bar is 2 μm). (**B**) Example traces of microtubule plus-end growth in the presence of GMPCPP at low (blue: 0.5 μM) and high (red: 1.5 μM) tubulin concentrations. Light traces show raw end positions at 10 frames/s, and darker lines represent a 10 frame (1 s) boxcar average. Kymographs are shown above the traces. Vertical scale bar is 10 min and horizontal scale bar is 1 μm. (**C**) Average growth rates vs. free GMPCPP-tubulin concentration. Error bars show the standard deviation of the growth rate at each concentration (n = 21, 59, 16, 51, 9, and 23 microtubules). The dotted line shows the linear fit, weighted by the inverse of the standard error of the mean (1/SEM). (**D**) Measurement of the growth variance for a single microtubule. Instantaneous displacements (ΔL) are recorded over different averaging windows, where n refers to the number of frames with exposure time $t_{exp}$ of 0.1 s averaged, and the variance of ΔL is plotted as a function of n (open black dots). The red line and shaded region show the fall in variance (approximately $1/\Delta t^1$) due to averaging out measurement error. The linear region denoted by the blue line and shading shows a crossover point where the measurement error becomes minimal and the increased variance is due to fluctuations around a mean growth rate. The black curve denotes the sum of these two phases. Reported values for growth variance in the paper refer to the 'D' term in the linear regime. (**E**) Average growth variance, resulting from fluctuations in microtubule growth, as a function of free tubulin. Error bars show the standard

*Figure 1 continued on next page*

*Figure 1 continued*

deviation of the measurement, and the linear fit is weighted by 1/SEM. A positive y-intercept is extrapolated back from the measurements at 1.2 nm²/s, which matches closely to our experimental noise floor measured on stationary taxol-stabilized GMPCPP microtubule seeds (gray open circle), 0.97 nm²/s.

The online version of this article includes the following source data and figure supplement(s) for figure 1:

**Source data 1.** Source data for *Figure 1C, E*.

**Figure supplement 1.** Growth variance significantly decreases with sub-pixel correction of microtubule growth.

**Figure supplement 1—source data 1.** Source data for *Figure 1—figure supplements 3 and 4*.

**Figure supplement 2.** Variations between microtubules growing under identical conditions.

**Figure supplement 3.** Mean displacement and variance (MD-Var) provides a better estimate for the growth variance than mean squared displacement (MSD).

**Figure supplement 4.** Transient tapers fluctuate around a constant value and occur more often at higher free tubulin concentrations.

that used MSD fitting to analyze fluorescence images (*Gardner et al., 2011*); likely causes of the discrepancy and implications of the smaller fluctuations for mechanisms of microtubule dynamics are addressed in the Discussion and *Figure 1—figure supplements 1–3*. The measurement noise in our assay was estimated to be 1.2 nm²/s by extrapolating the growth variance to zero tubulin concentration. Similarly, measuring the end position of a static GMPCPP and taxol-stabilized microtubule seed yielded a growth variance of 0.97 nm²/s (*Figure 1E* – gray point). This contribution from noise means that our measurements provide an upper limit on the fluctuations around the average microtubule growth rate – the actual fluctuations are likely smaller than the values we report.

We estimated the length of the microtubule end taper by quantifying the fall off in image intensity at the end of a growing microtubule (*Gardner et al., 2011*; *Maurer et al., 2014*). To calibrate our taper measurements, we used the known characteristics of our microscope to simulate images of synthetic microtubules with defined taper lengths and then fit those images to quantify the fall off in intensity at the end (*Figure 1—figure supplement 4*). The minimal taper length that was distinguishable above the point spread function of the microscope was 265 nm (~33 tubulin), similar to *Maurer et al., 2014*. We found that with increasing tubulin concentrations, the fraction of time that end tapers were detectable above the measurement threshold increased (*Figure 1—figure supplement 4*), and that tapers alternately grew and shrank, switching on timescales of tens of seconds (*Figure 1—figure supplement 4*).

## A minimal computational model can largely recapitulate measured growth rates and fluctuations

To obtain insight into the biochemistry underlying microtubule growth rates and fluctuations in the presence of GMPCPP, we applied a minimal kinetic model that simulates microtubule elongation at the level of individual association and dissociation events. This model, which has been described previously (*Ayaz et al., 2014*; *Kim and Rice, 2019*; *Mickolajczyk et al., 2019b*; *Piedra et al., 2016*) and which is similar to other biochemical models (*Gardner et al., 2011*; *Margolin et al., 2012*; *VanBuren et al., 2002*), is summarized in *Figure 2—figure supplement 1*. The two parameters that determine growth rate in the model (*Figure 2A*; *Figure 2—figure supplement 1*) are the strengths of longitudinal and 'corner' (one longitudinal + one lateral) interactions between αβ-tubulin and the lattice. The rate of αβ-tubulin:microtubule associations in the model is determined by the bimolecular association rate constant $k_{on}$ and the concentration of αβ-tubulin (via $k_{on}*[αβ\text{-tubulin}]$). Dissociation rates for different sites are calculated from $K_D$, the equilibrium dissociation constant for that site (via $k_{on}*K_D$). The experiments do not directly determine the association rate constant, so we tested a range of values when trying to fit the model to data. We also tested a number of different corner affinities ($K_D^{corner}$) for each $k_{on}$. For each combination of $k_{on}$ and $K_D^{corner}$, we optimized the longitudinal affinity ($K_D^{long}$) to obtain the best fit to experimental data.

We found that the αβ-tubulin on- and off-rate constants in GMPCPP cannot be uniquely constrained by fitting the microtubule growth rates alone. As long as the corner affinity is sufficiently high ($K_D^{corner}$ less than ~100 nM), the model can recapitulate the concentration-dependent growth rates in GMPCPP for assumed $k_{on}$ values spanning an ~10-fold range (from 0.27 to 2.2 µM⁻¹ s⁻¹) (*Figure 2—figure*

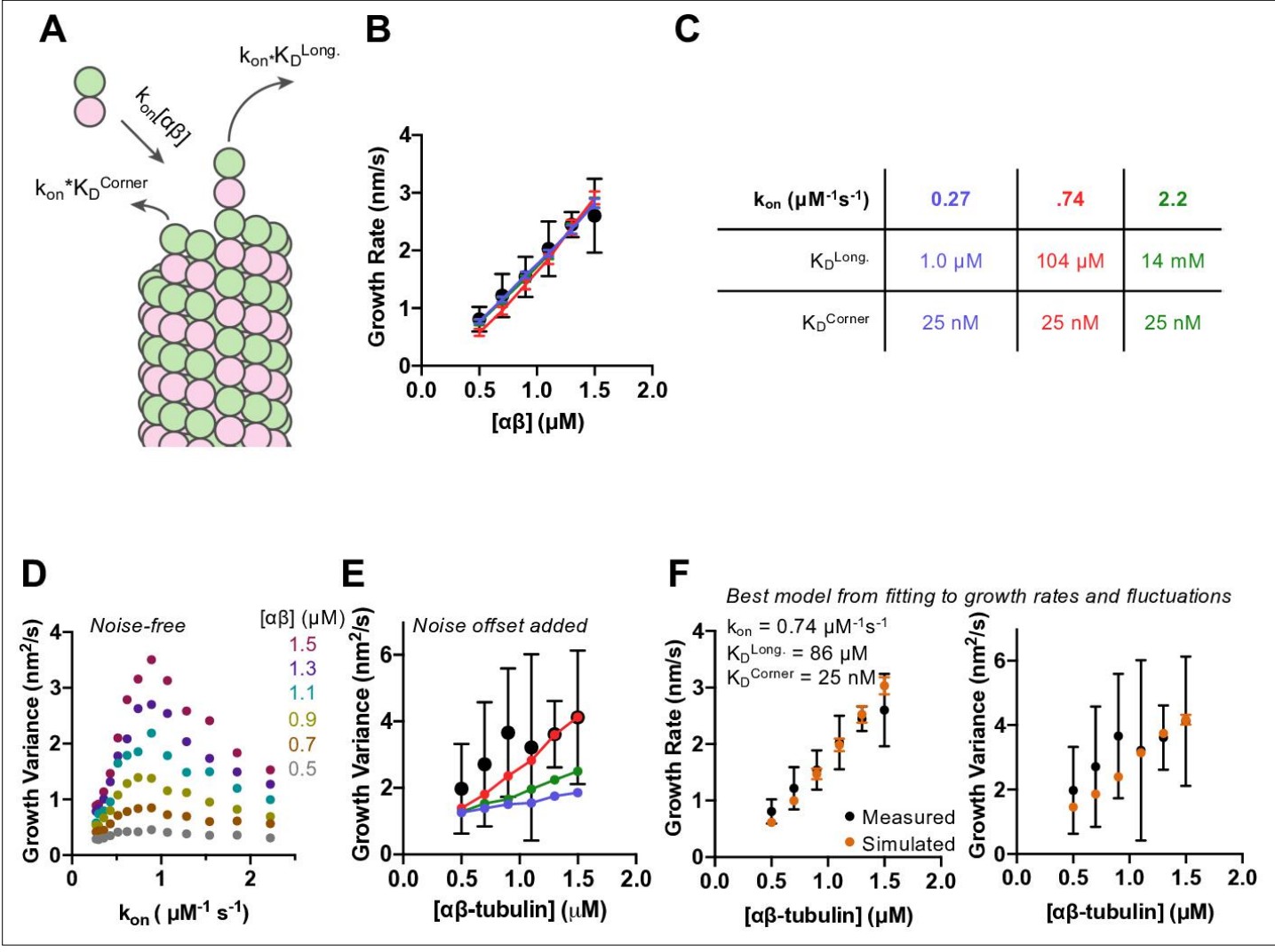

**Figure 2.** Simulations recapitulate GMPCPP growth rates for a wide range of association rates and longitudinal affinity, but fluctuations are only captured in a narrower range. (**A**) Cartoon showing the dominant factors that determine the concentration-dependent growth rates: association rate is determined as $k_{on}*[\alpha\beta]$, where $k_{on}$ is the on-rate constant, and dissociation rates from longitudinal (no lateral neighbors) and corner (one lateral neighbor) sites are determined as $k_{on}*K_D$, where $K_D$ represents the dissociation constant for the given type of site (see Materials and methods). (**B**) Measured (filled circles; data from *Figure 1*) and simulated growth rates (lines) after fitting the model to the measurements for three different assumed $k_{on}$ values (blue: 0.27 $\mu M^{-1}$ $s^{-1}$, red: 0.74 $\mu M^{-1}$ $s^{-1}$, green: 2.22 $\mu M^{-1}$ $s^{-1}$). Error bars show SD (n = 50) for experimental and simulated values; when not visible, it is because the bars are obscured by the symbol. (**C**) Fitted parameters for the three fits depicted in B. (**D**) Fluctuations around the average simulated growth rate as produced by fitting the model to the measurements for a larger set of assumed $k_{on}$ values than in B. The magnitude of fluctuations in simulations shows a bell-shaped dependence on $k_{on}$, reaching the highest values in the middle of the range explored. (**E**) Measured (filled black circles; data from *Figure 1*) and simulated (filled circles connected by lines; color coding as in B) fluctuations around the average growth rate. A 'noise offset' of 0.97 $nm^2$/s (see *Figure 1E*) has been added to the simulation data to facilitate comparison with the measurements. Error bars are as described in (**B**). (**F**) Result of fitting the model to growth rates (left) and fluctuations (right) simultaneously. Measured values are from *Figure 1* and depicted as filled circles; values from simulations are $k_{on}$: 0.74 $\mu M^{-1}$ $s^{-1}$, $K_D^{long}$: 86 $\mu M$, $K_D^{corner}$: 25 nM. Error bars show SD (n = 50).

The online version of this article includes the following video, source data, and figure supplement(s) for figure 2:

**Source data 1.** Source data for *Figure 2D-F*.

**Figure supplement 1.** Explanation and illustration of the model.

**Figure supplement 2.** Model fit to measured growth rates as a function of corner affinity.

**Figure supplement 3.** Differing magnitude variation in simulated microtubule growth at different association rate constants.

**Figure supplement 4.** The taper and roughness of microtubule end structures from simulations depends on the regime of association rate and longitudinal affinity.

*Figure 2 continued on next page*

*Figure 2 continued*

**Figure 2—video 1.** Fluctuations in microtubule growth rate and extent of end taper vary for different combinations of association rate and longitudinal affinity.

https://elifesciences.org/articles/75931/figures#fig2video1

*supplement 2*; selected conditions are shown in *Figure 2B*). Longitudinal interactions are predicted to be strongest ($K_D^{long}$ ~1 μM affinity or stronger) for the smallest $k_{on}$ value tested. Under conditions of high longitudinal affinity, pure longitudinal associations onto a protofilament are relatively long-lived and consequently most of them become incorporated into the growing microtubule. Conversely, longitudinal interactions are predicted to be weakest ($K_D^{long}$ ~10 mM affinity or weaker) for the highest $k_{on}$ value tested. Under conditions of low longitudinal affinity, most pure longitudinal associations onto a protofilament are relatively short-lived and do not contribute meaningfully to elongation. This inverse relationship between assumed $k_{on}$ and $K_D^{long}$ holds for intermediate choices of $k_{on}$ and yields different degrees of end tapering (*Figure 2—figure supplement 4*; *Ayaz et al., 2014*; *Cleary and Hancock, 2021*; *VanBuren et al., 2002*).

Different choices of $k_{on}$ (with their correspondingly different longitudinal affinities) yielded different magnitude fluctuations in growth rate that were evident in the relative divergence of length vs. time traces (*Figure 2—figure supplement 3*). Quantifying the fluctuations from 50 simulated growth episodes revealed a bell-shaped response in which intermediate values of $k_{on}$ produced the largest fluctuations in growth rate (*Figure 2D*). This variation in the magnitude of fluctuations as a function of $k_{on}$ suggests that the fluctuations are reflecting some property of the microtubule end that varies with longitudinal affinity. To compare the (noise-free) fluctuations obtained from the simulations to those measured in experiments, we added to the simulated values the 'noise offset' of 0.97 $nm^2/s$ obtained from measuring the fluctuations of GMPCPP seeds that were not detectably elongating. With this correction, the simulations using different combinations of $k_{on}$, $K_D^{corner}$, and $K_D^{long}$ produce fluctuations in growth rate that can be close to those measured experimentally (*Figure 2E*).

The model was able to recapitulate measured growth rates using a variety of assumed $k_{on}$ values. To better define the parameters required to reproduce the measurements, we used the observed growth rate fluctuations as an additional fitting constraint. The optimal parameters that emerged from this more constrained fitting are: $k_{on}$ = 0.74 $μM^{-1}$ $s^{-1}$, $K_D^{corner}$ = 25 nM, and $K_D^{long}$ = 87 μM (*Figure 2F*). There is some degeneracy: had we chosen a slightly stronger or weaker corner affinity; the associated longitudinal affinity would differ correspondingly. The parameters we obtained give rise to microtubule end structures that are somewhat tapered (*Figure 2—figure supplement 4*), but less tapered than observed experimentally (*Figure 1—figure supplement 4*).

## Fluctuations in microtubule growth rate are substantially increased in the presence of GTP

To determine whether and how GTP hydrolysis affected growth rates and fluctuations, we analyzed 'dynamic' microtubules growing with GTP. We measured growth rates and fluctuations as described above, but using higher concentrations of tubulin because sustained microtubule elongation with GTP did not occur below a tubulin concentration of ~6 μM. Representative kymographs and tracks of microtubule length as a function of time are shown in *Figure 3A* for the low (7.5 μM) and high (17.5 μM) concentrations used.

Average growth rates for the dynamic microtubules (*Figure 3B*) are roughly 10-fold greater than for the GMPCPP measurements (*Figure 1*), mirroring the higher tubulin concentration. A linear fit to the growth rates yields a $k_{on}^{app}$ of 2.3 $μM^{-1}$ $s^{-1}$, close to what we obtained for GMPCPP ($k_{on}^{app}$ of 3.1 $μM^{-1}$ $s^{-1}$, see *Figure 1C*). The apparent critical concentration (x-intercept) for microtubule growth of 3.9 μM with GTP is about 35-fold greater than for growth with GMPCPP (~50 nM, *Figure 1C*), reflecting weaker interactions of tubulin with the microtubule end in the presence of GTP. The measured growth rates and the differences between growth rates with GTP or GMPCPP are consistent with prior work (*Gardner et al., 2011*).

The fluctuations in growth rate for microtubules growing with GTP are several-fold larger than those obtained with GMPCPP, and depend ~3-fold more steeply on concentration (slope of 5.6 $nm^2/s/$μM for GTP, *Figure 3B*; slope of 1.9 $nm^2/s/$μM for GMPCPP, *Figure 1D*). Notably, the magnitude of fluctuations in GTP is ~7-fold smaller than reported in a prior study that used a different method to

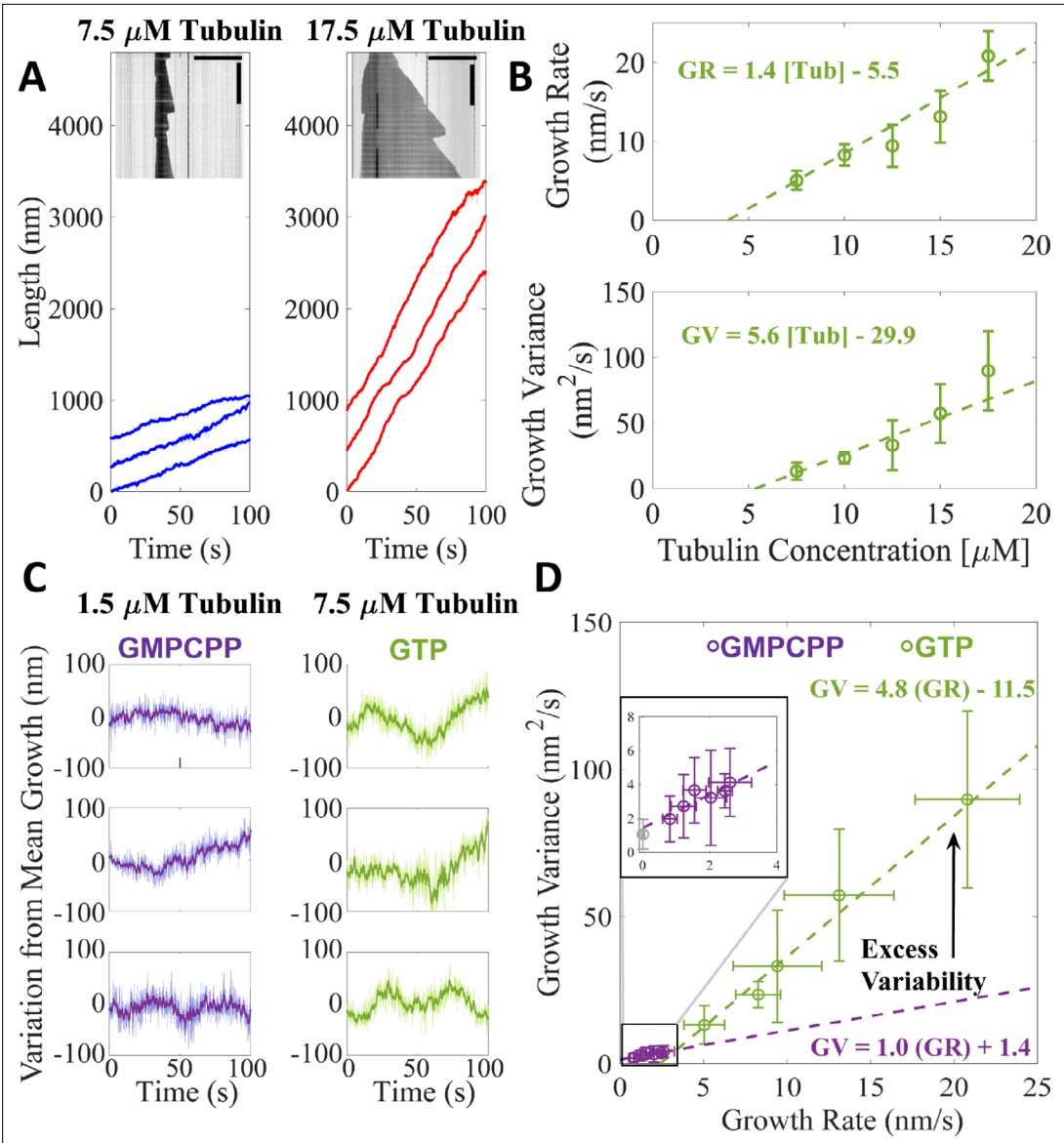

**Figure 3.** Microtubules growing in GTP show excess fluctuations compared to growth in GMPCPP. (**A**) Example tracks of microtubule growth in GTP at 7.5 µM free tubulin (left, blue) and 17.5 µM free tubulin (right, red). The light shading represents the tip traces acquired at 10 frames/s and the darker traces denote a 10-frame boxcar average of the traces over time. Example kymographs of dynamic microtubules are shown as insets; horizontal scale bar is 10 µm and the vertical scale bar is 5 min. (**B**) Average microtubule growth rate (top) and average growth variance (bottom) as a function of free tubulin concentration in GTP. Error bars show standard deviation (n = 34, 20, 16, 15, 13 growth events). A linear fit, weighted by 1/SEM, is shown as the dashed green line. (**C**) Deviations from expected growth length (given by the product of the average growth rate and time) for microtubules grown in GMPCPP (left – *Figure 1A*) and GTP (right – **A**). (**D**) Growth variance vs. growth rate for microtubules grown in GMPCPP (purple) and GTP (green). Error bars are SD. The inset more clearly shows the linear relationship between growth rate and growth variance in GMPCPP (purple). The experimental noise floor is shown as the gray point in the inset. Dashed lines denote linear fits weighted by 1/SEM of the growth variance.

The online version of this article includes the following source data and figure supplement(s) for figure 3:

**Source data 1.** Source data for *Figure 3B, C*.

**Figure supplement 1.** Long transient tapers occur during microtubule growth in GTP.

analyze fluctuations (*Gardner et al., 2011*), even though measured growth rates were comparable between the two studies. Deviations from expected mean growth are larger at 7.5 µM GTP-tubulin than at 1.5 µM GMPCPP-tubulin (*Figure 3C*), but because the growth rates in GTP are considerably faster than in GMPCPP, it is not clear how much the increased fluctuations reflect real differences in growth properties or simply the faster overall growth rates. The fluctuations around the average

microtubule growth rate at 1.5 µM GMPCPP-tubulin and 7.5 µM GTP-tubulin change more (3.2-fold, from 4.1 to 13.1 nm²/s) than the growth rate (2-fold, from 2.6 to 5.1 nm/s). To provide a better way to compare the magnitude of fluctuations between the two conditions, we plotted the measured fluctuations as a function of the corresponding microtubule growth rates (*Figure 3D*). Extrapolating the GMPCPP fluctuations out to faster growth rates reveals that microtubules growing with GTP fluctuate substantially more around their average growth rate than microtubules growing with GMPCPP, even accounting for different growth rates.

## In the computational model, transient protofilament poisoning by GDP can explain the increased growth fluctuations of dynamic microtubules

What is the biochemical origin of increased microtubule growth fluctuations in the presence of GTP? We used the computational model to begin addressing this question. We fit the model to the concentration-dependent GTP growth rates, initially ignoring GTP hydrolysis to mirror how we analyzed the GMPCPP data. Because the $k_{on}^{app}$ was similar for GTP and GMPCPP measurements, and in the absence of other justification to choose a different value, we used the optimal $k_{on}$ determined from the GMPCPP analysis (0.74 µM⁻¹ s⁻¹) and tested a range of corner affinities. The corner affinity that best fits the GTP growth rates was $K_D^{corner}$ = 2.9 µM (*Figure 4A*), roughly 100-fold weaker than for the GMPCPP data and consistent with the large change in apparent critical concentration. The longitudinal affinity that best fit the GTP growth rates was $K_D^{long}$ = 1.7 mM, also weaker than we obtained for GMPCPP.

While the 'GTPase-free' model was able to fit the observed growth rates, it dramatically underestimated both the magnitude of measured fluctuations and their concentration dependence (*Figure 4B*). The inability to recapitulate fluctuations is notable in light of the fact that the higher tubulin concentration (which gives a faster rate of association) and weaker corner affinity (which increases the rates of dissociation compared to the GMPCPP simulations) might have been expected by themselves to give much larger fluctuations. Altering the model (see Materials and methods) to use different on-rate constants for longitudinal, corner, and 'bucket' binding sites (*Castle and Odde, 2013*; *Gardner et al., 2011*) also did not improve predictions of fluctuations (*Figure 4—figure supplement 1*). Thus, in contrast to what we observed for elongation in GMPCPP, the GTPase-free model cannot recapitulate the growth fluctuations measured in GTP. Instead, some other mechanism or state that is missing from the minimal model must be needed to generate larger fluctuations.

We simulated microtubule dynamics in the presence of GTPase activity to test whether GTPase could represent this missing, fluctuation-increasing mechanism. We assume GTP acts in trans to stabilize the lattice because this is most consistent with biochemical (*Piedra et al., 2016*; *Rice et al., 2008*) and structural data (*Alushin et al., 2014*; *Zhang et al., 2015*). Incorporating GTPase activity required two additional parameters (*Kim and Rice, 2019*; *Piedra et al., 2016*; *VanBuren et al., 2002*; see Materials and methods): the rate constant for GTP hydrolysis and the weakening effect of GDP on the strength of tubulin:tubulin interactions (*Figure 4C*). Because of the potential degeneracy in parameter values mentioned above, we focused on trends with increasing GTPase rate rather than focusing on one particular value of the GTPase rate. As the rate of GTPase activity was increased in successive simulations, microtubule growth rates decreased and their fluctuations increased (*Figure 4D*). It is notable and somewhat counterintuitive that the fluctuations increased as growth rates decreased, because slower growth would normally be expected to produce smaller fluctuations (as seen in *Figure 3D*). The magnitude of the fluctuations increased sufficiently to match or even exceed the experimental measurements (dashed red line in *Figure 4D*). Thus, GTPase activity and its introduction of a second biochemical state allows the simulations to recapitulate the experimentally observed magnitude of fluctuations.

GTPase activity cannot occur in the terminal plus-end subunits because GTPase activity requires that a longitudinal interface be formed 'above' the GTP (*Nogales et al., 1998*). So how does GTPase activity affect microtubule growth rates and fluctuations, which are dictated by the binding and unbinding of subunits at the microtubule end? GDP-bound tubulins can be exposed on the microtubule end as a consequence of the dissociation of more terminal subunits (closer to the protofilament end). Instantaneous growth rates calculated from simulations without GTPase (*Figure 4E*, top left) vary in a relatively narrow range, whereas those calculated from simulations with GTPase (*Figure 4E*, top right) show much larger variations. In the simulations with GTPase, the largest decreases in growth

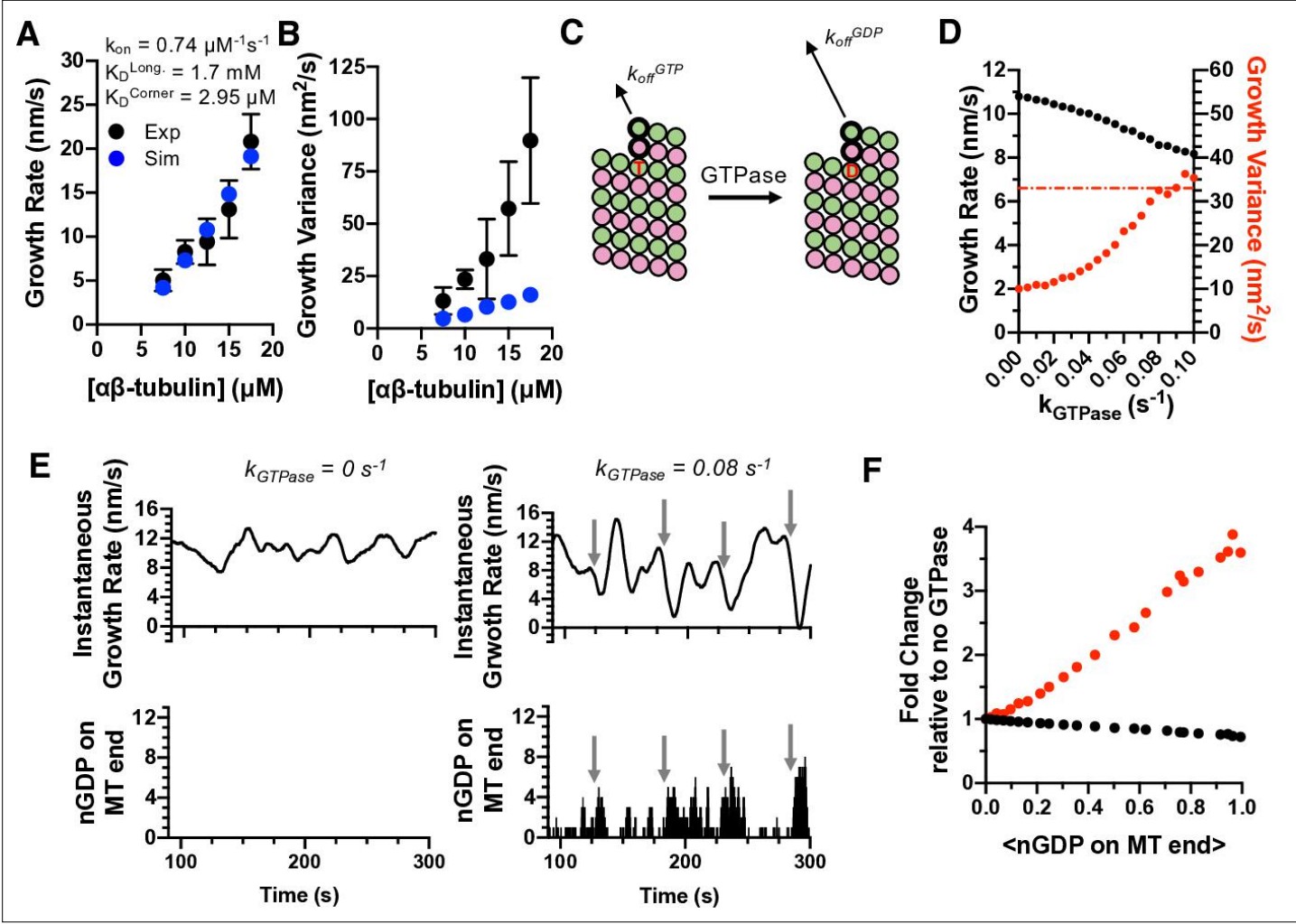

**Figure 4.** Exposure of GDP-tubulin on protofilament ends in simulations explains the excess fluctuations observed in GTP. (**A**) Simulated growth rates for microtubules in GTP (blue) provide a good match to measured values (black; data from **Figure 3**). The GTP simulations used the optimal on-rate constant determined for GMPCPP-tubulin ($k_{on}$: 0.74 $\mu M^{-1}$ $s^{-1}$; **Figure 2**) and used corner and longitudinal affinity values ($K_D^{corner}$ = 2.9 $\mu M$; $K_D^{long}$ = 1.7 mM) optimized to fit the experimental growth rates in GTP. Error bars show SD (n = 50) for experimental and simulated values; when not visible, it is because the bars are obscured by the symbol. (**B**) Fluctuations around the average growth rates in simulations (blue) substantially underestimate the measured values (black; data from **Figure 3**). Error bars are as in (**A**). (**C**) Cartoon illustrating how GTPase activity is incorporated into the model. Two additional parameters are required: a rate constant for GTP hydrolysis, and a scale factor to model the 'weakening' effect of GDP on the longitudinal interface. (**D**) Results from simulations at 12.5 $\mu M$ $\alpha\beta$-tubulin (interaction strengths as in panels A and B) including GTPase activity. As the rate of GTPase increases in simulations, growth rates (black) decrease, whereas fluctuations in growth rate (red) increase to the point that they are comparable to the measured value (dotted line). (**E**) Exposure of GDP on the growing end of the microtubule coincides with larger fluctuations in growth rate. In the absence of GTPase (left plots), the instantaneous growth rate varies in a relatively narrow range (top) and there is no GDP exposed on the microtubule end (bottom). When GTPase activity is present (right plots), the instantaneous growth rate explores a larger range including much slower values (top), and these especially slow growth rates coincide with exposure of one or more GDP-terminated protofilaments (bottom). (**F**) Dependence of growth rate (black) and growth fluctuations (red) on the average number of GDP exposed on the growing microtubule end. Values are plotted as fold change relative to the no GTPase condition. Fluctuations respond more strongly to the number of exposed GDP than do growth rates.

The online version of this article includes the following video, source data, and figure supplement(s) for figure 4:

**Source data 1.** Source data for **Figure 4A, B, D, F**.

**Figure supplement 1.** Site-dependent association rates do not explain the larger growth fluctuations observed in GTP.

**Figure supplement 1—source data 1.** Source data for **Figure 4—figure supplement 1C**.

**Figure 4—video 1.** GTP hydrolysis results in large fluctuations in microtubule growth.

https://elifesciences.org/articles/75931/figures#fig4video1

rate occur when multiple protofilaments have a terminal GDP (compare *Figure 4E* top and bottom). Even a relatively low average frequency of GDP exposure shows demonstrable effects on microtubule growth rates, and even more so on fluctuations (*Figure 4F*).

## Discussion

### Improved methods for measuring microtubule growth rates and fluctuations

We used IRM (*Mahamdeh and Howard, 2019*; *Mahamdeh et al., 2018*) to measure microtubule dynamics with high time resolution and without the need for fluorescent labeling, which allowed for longer recordings. The IRM imaging provided high signal to noise and allowed the position of the microtubule end to be defined with sub-pixel precision throughout relatively long (several minutes) time-lapse series. Whereas the growth rates we measured for GMPCPP and GTP are largely consistent with prior studies (*Brouhard et al., 2008*; *Gardner et al., 2011*; *Vemu et al., 2017*), the associated fluctuations were substantially smaller than previously reported (*Gardner et al., 2011*; *Rickman et al., 2017*). Several factors contribute to the reduced magnitude fluctuations we report here: sub-pixel drift correction, a more robust method for determining fluctuations, and the ability of IRM to image microtubule growth at higher frame rates and long durations. The impact of these lower magnitude fluctuations on constraining mechanisms of microtubule dynamics and regulation is discussed below.

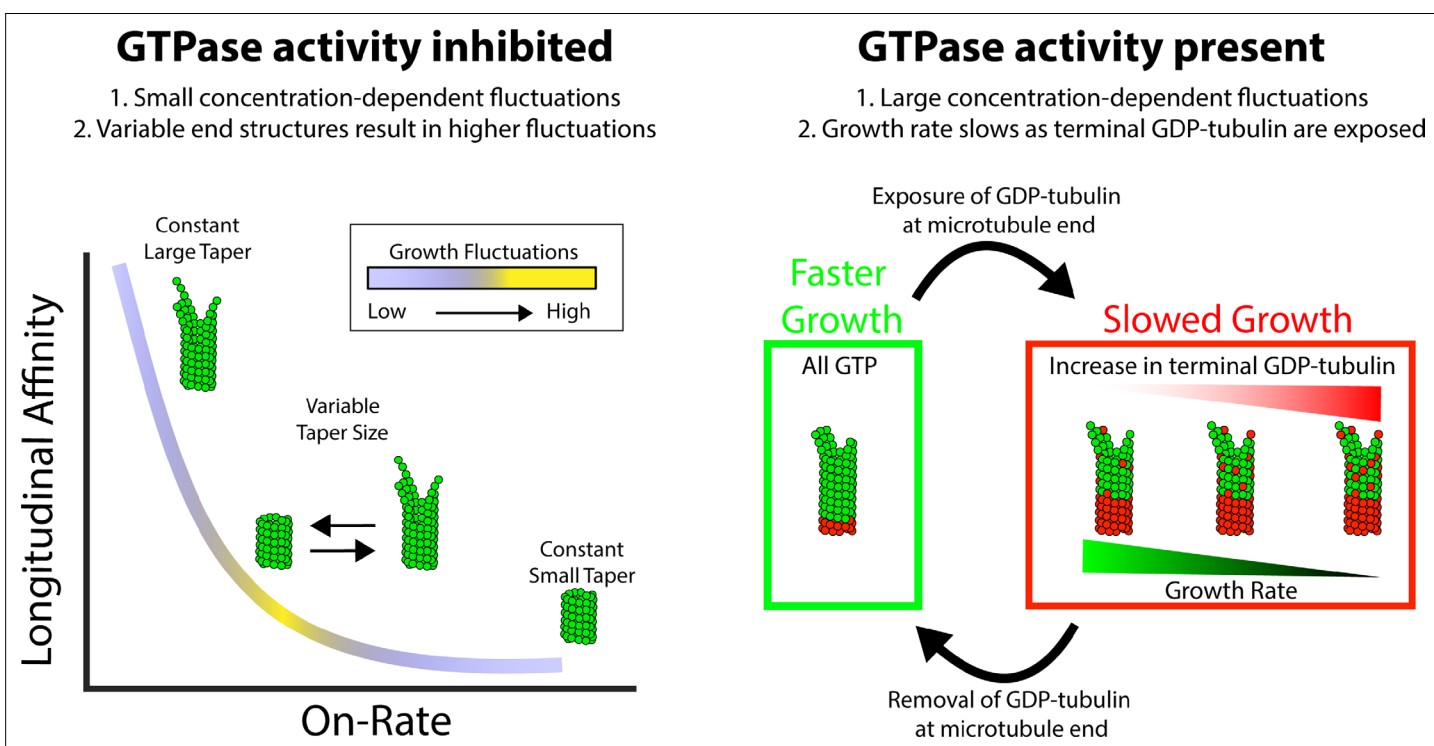

**Figure 5.** Summary of results and insights. (Left) Microtubule growth in the absence of GTPase activity can be accounted for by a simple biochemical model that incorporates the tubulin on-rate and the longitudinal affinity of the terminal tubulin at the plus-end. A slow on-rate and strong longitudinal affinity produce small fluctuations in growth and a large taper because the protofilaments grow somewhat independently of one another. A fast on-rate and weak longitudinal affinity produce small fluctuations in growth and a small taper because only corner interactions contribute to productive growth. The middle range produces the largest fluctuations in growth rate and best matches the measured fluctuations. The reason that this regime produces the largest fluctuations is because is where the average taper is itself varying the most, which leads to episodes of faster or slower growth depending on the end configuration. (Right) When GTPase activity is present, the growth fluctuations are much larger than in GMPCPP where hydrolysis is absent. The growth rate fluctuates more drastically due to switching between fast growth phases, when all terminal tubulins contain GTP, and slow growth phases resulting from one or more exposed GDP-tubulin at the tip. When the exposed GDP-tubulin either dissociates or is buried by an incoming GTP-tubulin, an all-GTP tip with a corresponding fast growth rate is restored.

## Microtubule elongation in the absence of GTPase: slow associations and small fluctuations

Microtubules grew very steadily with only small fluctuations in growth rate when GTPase activity was suppressed. Kinetic simulations implementing a minimal biochemical model for microtubule elongation recapitulated the observed growth rates using a range of association rate constants, with slower association rates compensated for by stronger longitudinal interactions. Our model is similar to that described in *VanBuren et al., 2002*, but with trans-acting GTP (*Piedra et al., 2016*; *Rice et al., 2008*). For simplicity, the model used in the present study does not incorporate additional states (such as GDP.Pi) or beyond-nearest neighbor effects (to mimic allostery in the lattice) (*Kim and Rice, 2019*); it also does not incorporate mechanochemical effects that attempt to capture 'spring-like' properties of the αβ-tubulin conformational change (*McIntosh et al., 2018*; *VanBuren et al., 2005*; *Zakharov et al., 2015*).

Fluctuations around the average growth rate in the model provided stronger fitting constraints because they were more sensitive to the choice of association rate constant: at comparable growth rates, fluctuations were highest around the middle of the range of association rate constants tested, and lower at the 'fast' and 'slow' extremes. The different characteristic magnitudes of fluctuations in growth rate reflected the extent to which the microtubule end configuration (number of longitudinal and corner sites) was itself varying in response to the different strength longitudinal affinities (*Figure 5*).

In our minimal model for microtubule dynamics, the parameters that gave the best fit to the GMPCPP growth rates and growth rate fluctuations corresponded to relatively slow associations and relatively strong longitudinal associations. The slower, stronger longitudinal interaction model that best describes our GMPCPP data is consistent with a recent single-molecule study of yeast microtubule elongation (*Mickolajczyk et al., 2019b*). It is also consistent with the emerging view (*Erickson, 2019*; *Gudimchuk and McIntosh, 2021*; *McIntosh et al., 2018*) that microtubule protofilaments can elongate somewhat independently of each other. This view is based on cryo-ET images showing flared plus-ends (*McIntosh et al., 2018*) and the associated mechanochemical models, and implicitly requires relatively strong longitudinal interactions.

## Transient exposure of GDP-terminated protofilaments: a major contributor to fluctuations in microtubule growth rate

Our measurements of microtubule elongation with GTP ('dynamic microtubules') showed fluctuations that were larger in magnitude than expected from extrapolation of the GMPCPP data. To fit the observed growth rates with a GTPase-free kinetic model required weaker longitudinal and corner binding affinities compared to the GMPCPP data, reflecting the more elevated critical concentration of the dynamic microtubules (growing in GTP), and revealing a significant biochemical difference between GMPCPP and GTP in the interactions that drive elongation. The affinities that best fit observed growth rates in the present work are in the same general range ($K_D^{corner} \sim 3$ μM, $K_D^{long} \sim 2$ mM) as those obtained fitting similar models to different datasets (e.g. *Kim and Rice, 2019*). The GTPase-free model, which only considers the GTP state of tubulin, could not recapitulate the growth fluctuations of dynamic microtubules. Because the GTPase-free model could recapitulate both the growth rates and fluctuations of microtubules growing in GMPCPP, it seemed likely that the fluctuations of dynamic microtubules reflect contributions from at least one additional biochemical state. Our modeling showed that exposure of GDP-bound αβ-tubulin on the end of protofilaments increases the variability in microtubule growth by transiently 'poisoning' elongation, yielding fluctuations comparable to those observed experimentally. Thus, the fluctuations of growing dynamic microtubules do not solely reflect properties of GTP-tubulin. The notion that growing microtubule ends expose GDP is not widely accepted, but we argue that it should be considered as an important factor. Indeed, the decrease in growth rate that accompanies GDP exposure(s) may represent an early event in the initiation of catastrophe (*Piedra et al., 2016*).

## Limitations of the modeling

We chose to use biochemically simplified models in the hopes of identifying the minimal set of parameters capable of recapitulating a given set of observations. Consequently, we did not attempt to explicitly account for either the assembly-dependent conformational changes that accompany

αβ-tubulin incorporation into the microtubule and that probably alter the strength of lattice contacts (*Brouhard and Rice, 2014*; *Brouhard and Rice, 2018*), or for 'mechanochemical' effects that consider spring-like properties of αβ-tubulin conformation (*Brouhard and Rice, 2018*; *Gudimchuk and McIntosh, 2021*). Incorporation of these different conformational states and/or transitions may improve the model's ability to simultaneously recapitulate growth rates, growth rate fluctuations, and end tapers. Because we refine the longitudinal and corner affinities independently (without imposing a fixed relationship between the two), some of these effects might be captured implicitly. We also did not allow for the possibility of GDP to GTP exchange on the microtubule end (*Piedra et al., 2016*). This is a lesser limitation because a finite rate of exchange would only modulate the amount of GDP on the microtubule end for a given GTPase rate; it would not eliminate the 'poisoning' effect of GDP exposure that increases fluctuations in growth rate. Whatever the limitations of our model, our results clearly show that (i) there are 'excess' fluctuations for microtubules with GTP compared to GMPCPP and (ii) a simple 'one-state' model can recapitulate the growth rates and fluctuations in GMPCPP, but an additional state is required to generate the larger fluctuations observed with GTP.

## Implications for mechanisms of dynamics and regulation

Our findings have three general implications for the mechanisms of microtubule dynamics and regulation. First, they support a model in which the association kinetics are relatively slow, with longitudinal associations correspondingly strong. This view is consistent with a recent single-molecule study of yeast microtubule growth (*Mickolajczyk et al., 2019b*), and it also resonates with structure-based evidence that microtubule protofilaments can elongate somewhat independently from one another (so-called flared ends) (*Erickson, 2019*; *Gudimchuk and McIntosh, 2021*; *McIntosh et al., 2018*). Second, although GDP-tubulin is not widely thought to be present on growing microtubule ends, our results indicate that transient exposure of GDP-tubulin on protofilament ends causes larger fluctuations in microtubule growth rate than would be expected from 'all GTP' elongation. Because the size of the GTP cap decreases with slower growth rates (*Duellberg et al., 2016*), this consequence for GDP-terminated protofilaments provides a possible mechanism for the initiating events of microtubule catastrophe. Finally, our findings provide new ways of thinking about potential mechanisms that regulatory factors use to control microtubule dynamics. The relatively slow association kinetics we observe imply that modulating the rate of αβ-tubulin association or dissociation each provide viable strategies for controlling microtubule elongation rates. Indeed, Stu2/XMAP215 family polymerases are thought to accelerate microtubule growth by increasing the rate of αβ-tubulin associations (*Ayaz et al., 2014*; *Ayaz et al., 2012*; *Brouhard et al., 2008*). More speculatively, it seems plausible that a regulatory factor might be able to control microtubule elongation or catastrophe by altering the lifetime of GDP-tubulin on the microtubule end, either by altering the rate of nucleotide hydrolysis or exchange, or by altering the rate at which the GDP-bound subunits dissociate from the growing end.

## Materials and methods
### Tubulin purification and labeling

PC-grade bovine brain tubulin was purified as previously described (*Uppalapati et al., 2009*), cycled twice, quantified by $A_{280}$ (ε = 115,000 $M^{-1}$ $cm^{-1}$), diluted to either 80 or 15 µM in BRB80 (80 mM K-Pipes, 2 mM EGTA, 2 mM $MgCl_2$, pH 6.9), aliquoted, and flash frozen in liquid nitrogen. Aliquots were stored at –80°C and used within 2 months.

Tubulin was biotinylated by polymerizing a solution of 40 µM free tubulin, 1 mM GTP, 4 mM $MgCl_2$, and 5% DMSO for 30 min at 37°C, adding an equimolar amount of EZ-Link NHS-Biotin in DMSO (ThermoFisher 20217), and reacting for 30 min at 37°C. The microtubules were then pelleted (all pelleting was 10 min at 30 psi in a Beckman Airfuge), the pellet was resuspended in cold BRB80 and incubated on ice for 30 min, spun for 10 min at 30 psi in a pre-cooled rotor (4°C), and the supernatant collected. The biotinylated tubulin was then subjected to another round of polymerization, pelleting, and depolymerization, and the concentration was checked using $A_{280}$. The fraction of biotin-labeled tubulin was measured using a Biocytin Biotin Quantification Kit (Thermo Scientific #44610). Final stocks were diluted using unlabeled tubulin to 40 µM with a 33% biotin-labeled fraction.

To make biotinylated microtubule seeds, 20 µM biotinylated tubulin was mixed with 1 mM GMPCPP and 4 mM $MgCl_2$, and incubated at 37°C for 60 min to form stable microtubule seeds. The solution

was then diluted to 2 µM tubulin with 0.5 mM GMPCPP and 2 mM MgCl$_2$ and allowed to grow at 37°C for 5 hr to generate longer seeds. The seeds were pelleted and resuspended in BRB80 with 20% glycerol, flash frozen in liquid nitrogen, and stored for up to 1 month at –80°C. On the day of the experiments, an aliquot of seeds was thawed at 37°C in the water bath, pelleted to remove the glycerol, resuspended in BRB80 with 0.5 mM MgCl$_2$ and 0.5 mM GMPCPP, and diluted as needed.

## Microtubule dynamics experiments

Cover slips (18 × 18 mm$^2$ Corning) were cleaned overnight in 6 M HCl, rinsed with ddH$_2$O, plasma cleaned (Harrick Plasma) for 12 min, and incubated in a vacuum-sealed desiccator with 1$H$,1$H$,2$H$,2$H$-perfluorodecyltrichlorosilane (Alfa Aesar L165804-03) overnight. Before use, the silanization was checked using a droplet test to confirm hydrophobicity.

Flow cells were made by fixing two pieces of double-sided tape about 10 mm apart to an ethanol washed and ddH$_2$O rinsed microscope slide, and covering with a silanized cover slip. 600 nM neutra-vidin (ThermoFisher) was flowed into the chamber, followed by 5% F127 (Sigma P2443-250G), 2 mg/mL casein (Sigma C-7078), and biotinylated microtubule seeds at a concentration to achieve 5–10 seeds per each 60 × 60 µm$^2$ field of view. Biotinylated BSA (1 mg/mL) was then added to the flow chamber to block any of the unused neutravidin on the cover slip. Finally, polymerization solution was added to the chamber, consisting of 0.05% methylcellulose (Sigma M0512-100G), 1 mM Mg-GTP or Mg-GMPCPP (Jena Biosciences), and an oxygen scavenging system (80 µg/mL Catalase [Sigma C1345-1G], 100 mM DTT, 200 mM D-Glucose [EMD Millipore Corp DX0145-1], 200 µg/mL Glucose Oxidase [EMD Millipore Corp 345386–10gm] in BRB80). The flow cell was then sealed using nail polish and let to sit for 5 min, and then placed onto the objective of the Nikon TE-2000 TIRF microscope that was pre-warmed using an objective heater to 30°C. Once the surface was found, the chamber was allowed to equilibrate to temperature for 5 min.

Microtubules were visualized using IRM (*Mahamdeh et al., 2018*) using a blue (440 nm) LED (pE-300white, CoolLED, UK) at 1.25% power. The LED was attached to the fluorescent line and the illumination NA was optimized to achieve the maximum signal to noise as shown in *Mahamdeh et al., 2018*. Videos were collected at 10 frame/s for 900 s.

## Video post-processing

Each video was flat-fielded to correct for uneven illumination, as follows. An out-of-focus image stack was acquired, the images averaged and converted to a 32-bit image in ImageJ (*Schneider et al., 2012*), and then normalized to a mean of 1 by dividing the intensity at each pixel by the average pixel intensity across the image. Flat fielding was achieved by dividing every frame of acquired videos by this background image. The flat-fielded image stacks were then inverted in ImageJ to generate light microtubules on a dark background, and converted to 8-bit images for analysis.

Stage drift was corrected by tracking fiduciary marks (small debris) found on the surface. Initially, 10 fiduciary marks were tracked using FIESTA (*Ruhnow et al., 2011*), and the tracks averaged. A custom-MATLAB script was built to correct for drift to the precision of one pixel. The script created a mask and centered the fiduciary marks to a given pixel on the mask throughout the movie. This pixel-corrected movie was used in subsequent analysis.

## Tracking microtubule growth

Microtubule growth was measured using a previously described tracking algorithm (*Demchouk et al., 2011*; *Prahl et al., 2014*) in MATLAB. In short, in the first frame of a movie, the user defines one point in the microtubule backbone and one at the microtubule tip. From these points, the software defines a region of interest around the microtubule and uses a centroid calculation to define the center point of the microtubule. Using pixel coordinates x' and y', for each pixel in x', the sub-pixel microtubule position in y' is determined by fitting the intensity I(y') to a Gaussian:

$$I\left(y^{'}\right) = I_{\text{Background}} + I_{\text{MT}}e^{-\frac{\left(y^{'}-y_{\text{CoM}}\right)^2}{2\sigma^2}} \tag{1}$$

where $I_{\text{Background}}$ is the mean background intensity, $I_{\text{MT}}$ is the peak intensity above background, $y_{\text{CoM}}$ is the center of mass of the microtubule backbone, and σ is the Gaussian standard deviation. Once the backbone positions are defined in the x' – y' coordinate system, a line is fit to the backbone and used

to create a new 1D coordinate system, x″, along the microtubule backbone. In x″, points are defined every 58 nm (the pixel size in x–y), and an intensity value, I(x″) is obtained by averaging the intensities of the closest five pixels in y. For each movie frame, the position of the microtubule tip, $\mu_{MT}$, is calculated by fitting a Gaussian survival function:

$$I\left(x^{''}\right) = \tfrac{1}{2}\,I_{MT}\,erfc\,\left(\tfrac{x^{''}-\mu_{MT}}{\sqrt{2}\sigma_{MT}}\right) + I_{Background} \tag{2}$$

where $I_{Background}$ is the mean background intensity, $I_{MT}$ is the peak signal above the background, and $\sigma_{MT}$ is the spread of the survival function. Using the defined line of the microtubule backbone, this tip position is then converted back to a sub-pixel tip position $\mu_{MT}(x',y')$ in the image plane.

We found that sub-pixel stage drift contributed to the apparent microtubule tip fluctuations, particularly for slowly growing microtubules, so we wrote a MATLAB code to remove this artifact. Sub-pixel drift was quantified by tracking the position of 10 fiduciary marks using FIESTA (*Ruhnow et al., 2011*) through the movie, and calculating their average position, Fid(x′, y′). Sub-pixel drift caused artifacts because the microtubule length is defined as the tip position $\mu_{MT}(x',y')$ relative a central backbone position B(x,y). Because B(x,y) is defined in the first frame, drift introduces an undetected error. This sub-pixel stage drift was corrected in two stages. Because the x position of the backbone was fixed by the pixel position, subtracting the fiduciary alone corrected for sub-pixel drift in x. Thus, the corrected microtubule length relative to the central backbone position was:

$$\Delta L_x = \mu_{\mathrm{MT}}\left(x\right) - B\left(x\right) - Fid\left(x\right) \tag{3}$$

The backbone position in y′ was determined by fitting I(y′) in every frame, such that sub-pixel movements in y were captured by the sub-pixel fit to the backbone (*Equation 1*). Thus, Fid(y) did not need to be considered. However, because the microtubule was at an angle ($\Theta$) to the x-axis and the x-position was set by the pixel position, sub-pixel drift in the x-direction led to an error in B(y). This necessitated the following correction:

$$\Delta L_y = \mu_{\mathrm{MT}}\left(y\right) - B\left(y\right) - \sin\Theta * Fid\left(x\right) \tag{4}$$

After these corrections, microtubule growth was calculated by the change of length in x and y:

$$\Delta L = \sqrt{\Delta L_x^2 + \Delta L_y^2} \tag{5}$$

## Analysis of microtubule growth

The tip displacement of each growing microtubule was tracked at 10 frames/s and sections of the tracks were selected that included a minimum of 90 s (900 points) of sustained growth with no tracking errors. Tracking errors were easily identified as large deviations from the constant growth, often for multiple frames in succession.

Tip growth fluctuations were analyzed following a method developed by *Castle et al., 2019* as follows. Microtubule tip displacement traces were downsampled to different effective frame rates by taking the mean tip position of n frames. This process was repeated from n = 2 to 300 points (lag time = 0.2–30 s), which created n−1 downsampled traces for each n. At each effective frame rate, instantaneous length displacement distributions (ΔL) were collected by subtracting the position in each 'frame' from the position in the previous 'frame'. For each lag time, the mean displacement, $\mu_{\Delta L}$, and the displacement variance, $\sigma_{\Delta L}^2$, were calculated by fitting a normal to the ΔL distribution.

The growth rate was calculated by fitting a line to the mean displacement at varying lag times using:

$$Growth\ Rate = \frac{\mu_{\Delta L}}{\Delta t} = \frac{\mu_{\Delta L}}{n * t_{\exp}} \tag{6}$$

where $t_{exp}$ is the original exposure time at 10 frames/s (0.1 s) and n is the size of the downsampling window.

The growth variance (D) was calculated by fitting the variance of the displacement at varying lag times, $\Delta t$, to a biphasic fit:

$$Growth\ Variance\ =\ 2D\Delta t + \frac{\sigma_i^2}{\Delta t^l} \tag{7}$$

At short lag times, the growth variance is dominated by experimental measurement error, $\sigma_i$, arising from fitting the position at the microtubule tip. This measurement error term, $\sigma_i$, decays at approximately $1/\Delta t^1$. The growth variance, D, is modeled as 1D diffusion and is calculated by fitting a line to the growth variance curve using a minimum lag time of 10 s to allow for both phases to be measurable. The best fit was determined by the $R^2$ value of the fit in MATLAB.

## Model convolution of microtubule end structure

We used model convolution (*Gardner et al., 2010*) to generate simulated microtubule images used for benchmarking the tip taper analysis. Briefly, microtubule backbones were generated, with an average size of 6.5 µm, following a 3-start helix B-lattice model. The radius of the microtubule was set to 25 nm and the tubulin dimer length was set to 8.2 nm. Tapered microtubule plus-ends were designed as a uniform distribution of protofilament lengths with stepwise increases in length occurring linearly from the 1st and 13th protofilament, making the center protofilament the longest. The average photon emission per tubulin subunit was determined by measuring the magnitude of the microtubule signal (1160 arbitrary intensity units, corresponding to a 5.2% contrast signal) relative to the average signal from back reflected light (22,300 units) on a 16-bit image (65,535 units maximum) (*Mickolajczyk et al., 2019a*). A mask was defined and the microtubule placed randomly along a 1024 × 1024 pixel image (59.4 × 59.4 µm²). Then photons were generated and distributed to various positions along the artificial image according to the Gaussian standard deviation calculated from the full-width half maximum (250 ± 90 nm) measured using the cross-section of the microtubule backbones in the images. Photon shot noise was then added based on the standard deviation of the background (181 units out of 65,535; this corresponds to a signal to noise of 6.2). The final image was then inverted and converted to an 8-bit image following the post-processing procedure used on the experimental videos and the microtubule tip was tracked using the same tracking algorithm (*Demchouk et al., 2011*; *Prahl et al., 2014*) used for the experimental measurements.

## Simulating microtubule growth

We used the simulation code and analysis algorithms described previously (*Kim and Rice, 2019*; *Mickolajczyk et al., 2019b*; *Piedra et al., 2016*), with minor modifications. Briefly (diagrammed in *Figure 1—figure supplement 1*), the code performs kinetic Monte Carlo simulations of microtubule polymerization dynamics. The overall approach is similar to *VanBuren et al., 2002*, with some differences described below. Microtubule dynamics are simulated one biochemical reaction at a time (αβ-tubulin subunit association or dissociation, and GTP hydrolysis), with the microtubule lattice represented by a 2D array with a periodic boundary condition that mimics the cylindrical wall and 'seam' of 13 protofilament microtubules. Association reactions are allowed at the end of each protofilament, and assumed to occur at a rate given by $k_{on}$[αβ-tubulin], where $k_{on}$ denotes the assumed on rate constant (but see below for a site-dependent association rate constants). Dissociation reactions occur at a rate given by $k_{on} \times K_D$, where $K_D$ is the affinity determined by the sum of longitudinal and lateral contacts for the dissociating subunit. In practice, this means that dissociations occur almost exclusively from terminal subunits. As described previously, (i) we do not allow pure lateral associations, and consequently the model parameter describing the strength of lateral contacts does not account for the entropic cost of subunit immobilization, and (ii) we do not allow completely surrounded subunits to dissociate. The GTPase reaction is assumed to occur at a rate given by $k_{GTPase}$ on all non-terminal subunits. Our parameterization assumes that the nucleotide (GTP or GDP) acts in trans (*Piedra et al., 2016*; *Rice et al., 2008*) to modulate the strength of longitudinal contacts. The modulation is represented as a multiplicative factor X such that $K_D(GDP) = X*K_D(GTP)$, where X is a large, positive number, and $K_D(GDP)$ and $K_D(GTP)$ represent the affinity of a given interaction with GDP or GTP, respectively, at the longitudinal interface. An 'execution time' for each possible event is chosen by randomly sampling an exponential distribution $e^{-rate*t}$, where rate denotes the characteristic rate for that type of event. At each step of the simulation, the fastest event (shortest execution time) is implemented, meaning that the microtubule configuration is updated, the list of possible events and rates refreshed accordingly, and the simulation time advanced by the execution time. In this way, different kinds of reactions occur with relative probability determined by their relative rates. The simulations output lists of user-specified

quantities (such as the number of tubulins in the microtubule, or the lengths of each protofilament, or the number of GTP-tubulins in the microtubule, or others) as a function of time. Code is available as a GitLab repository: https://git.biohpc.swmed.edu/s422146/simulate-mt-44 (*Rice, 2022*).

## Parameter estimation

We used iterative fitting in MATLAB to identify model parameters that could recapitulate observed measurements of growth rates or growth rate fluctuations. Briefly, 50 independent simulations of length 600 s were run for a given set of parameters and using the same concentrations as for the measurements (GMPCPP: 0.5, 0.7, 1.1, 1.3, and 1.5 μM; GTP: 7.5, 10.0, 12.5, 15.0, and 17.5 μM). The discrepancy between chosen simulation outputs (normally growth rates but in some cases also fluctuations in growth rate), averaged over the independent simulations, was minimized by varying the longitudinal and corner affinities. For simulations that included GTPase activity, we first identified the 'GDP weakening factor' that gave shrinking rates for all GDP microtubules that were comparable to the measured shrinking rate.

## Allowing site-dependent on-rate constants

We modified the model to allow for different on-rate constants for different kinds of binding sites (longitudinal, corner, or bucket sites, which correspond to a longitudinal contact with 0, 1, or 2 lateral contacts). Site-dependent on-rate constants were previously described in *Castle and Odde, 2013*, and we used similar ratios between longitudinal, corner, and bucket on-rate constants identified in that work ($k_{on}^{long}$ / $k_{on}^{corner}$ = 1.67, $k_{on}^{corner}$ / $k_{on}^{bucket}$ = 4.62). We set the association rate constant into corner sites to be identical to that used in the simpler, single association rate constant simulations, and then scaled the longitudinal and bucket associations to be faster and slower, respectively. This way of doing things minimized the differences in the strength of longitudinal and corner interactions between the single and multiple association rate constant simulations. Fitting the simulations to the measurements was performed as described above.

## Acknowledgements

This study was supported by NSF MCB-1615938 and NIH R01-GM135565 to LMR, and by NIH R35 GM139568 to WOH. JC received support from NIH T32 GM108563, and TK received support from NIH T32 GM008297. We thank D Odde for helpful discussions on best practices for determining growth rate fluctuations.

---

## Additional information

### Funding

| Funder | Grant reference number | Author |
|---|---|---|
| National Science Foundation | MCB-1615938 | Tae Kim<br>Lauren A McCormick<br>Luke M Rice |
| National Institutes of Health | R01-GM135565 | Lauren A McCormick<br>Luke M Rice |
| National Institutes of Health | R35-GM139568 | Joseph M Cleary<br>William O Hancock |
| National Institutes of Health | T32-GM108563 | Joseph M Cleary |
| National Institutes of Health | T32-GM008297 | Tae Kim |

The funders had no role in study design, data collection and interpretation, or the decision to submit the work for publication.

## Author contributions
Joseph M Cleary, Conceptualization, Formal analysis, Investigation, Methodology, Software, Validation, Writing - original draft, Writing - review and editing; Tae Kim, Conceptualization, Formal analysis, Methodology, Software, Validation, Writing - review and editing; Annan SI Cook, Formal analysis, Methodology, Software; Lauren A McCormick, Resources, Software, Validation; William O Hancock, Conceptualization, Funding acquisition, Project administration, Supervision, Writing - review and editing; Luke M Rice, Conceptualization, Formal analysis, Funding acquisition, Investigation, Methodology, Project administration, Supervision, Validation, Writing - original draft, Writing - review and editing

## Author ORCIDs
Lauren A McCormick ⓘD http://orcid.org/0000-0001-9164-0932
William O Hancock ⓘD http://orcid.org/0000-0001-5547-8755
Luke M Rice ⓘD http://orcid.org/0000-0001-6551-3307

## Decision letter and Author response
Decision letter https://doi.org/10.7554/eLife.75931.sa1
Author response https://doi.org/10.7554/eLife.75931.sa2

# Additional files

## Supplementary files
• Transparent reporting form

## Data availability
Numerical data used to generate most of the figures have been provided as Source Data.

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
