## [Editor Report]

Using improved optical imaging of growing microtubules and improved data analysis, the authors find that microtubule growth fluctuations are less pronounced than previously reported and that GTP hydrolysis in microtubules increases these fluctuations. A mathematical model of microtubule growth suggests that occasionally exposed GDP at the microtubule end can be responsible for increased growth fluctuations. This work provides new mechanistic insight into the molecular mechanism of microtubule growth.

---

## [Decision Letter]

**Decision letter after peer review:**

Thank you for submitting your article "Measurements and simulations of microtubule growth imply strong longitudinal interactions and reveal a role for GDP on the elongating end" for consideration by *eLife*. Your article has been reviewed by 3 peer reviewers, one of whom is a member of our Board of Reviewing Editors, and the evaluation has been overseen by Anna Akhmanova as the Senior Editor. The following individual involved in review of your submission has agreed to reveal their identity: Alex Mogilner (Reviewer #2).

Essential revisions:

1) Please provide more context and a broader motivation for the study in the Introduction making sure relevant literature is cited and shorten the summary of the results in the Introduction to avoid repetition.

2) Model: Please explain if/why a 2D model is needed for microtubule growth (instead of only single filament model).

3) Discussion: Please compare the model and the parameters used here to the relevant previous modelling studies, including a recent study of the authors (see reviewers' comments).

4) Please make sure that all error bars are provided and clearly stated.

5) Please make other editorial changes where appropriate to improve clarity of presentation, considering the suggestions made by the reviewers.

*Reviewer #1 (Recommendations for the authors):*

1. The Introduction currently provides a long summary of the results after a rather short introduction of previous work addressing the question of microtubule growth fluctuations. The topic could be introduced in a more comprehensive and a more conceptual manner and the summary of the results could in turn be shortened.

2. The Discussion could more clearly explain the difference between previous exclusively taper-shape dependent growth models and the model here which is a mixture of a taper shape and GTP hydrolysis model. Particularly why previous taper shape-dependent models that could explain even larger growth fluctuations now seem to fail for the reduced growth fluctuations of GTP microtubules as measured here.

*Reviewer #2 (Recommendations for the authors):*

I am not a serious experimental expert, but from my limited perspective the experiments are done very well. The model, simulations and comparison with the data are flawless.

*Reviewer #3 (Recommendations for the authors):*

The manuscript is rough and needs significant revisions:

– The Introduction devotes too much text to summarizing the results (all of page 3 basically) and too little text to introducing the question and the state of the debate. It's really only the 2nd paragraph that introduces pertinent papers. The authors should reach further back than Gardner 2011, because Gardner 2011 was itself part of a series of studies whose intent was to measure the kinetics of polymerization. E.g. Kerssemakers et al., Nature 2006 and Schek et al., Curr Biol 2007 come to mind. Classic citations relating to association kinetics, like Northrup and Erickson PNAS 1992, would help contextualize the tubulin results in a broader biochemical context. As it stands, the motivation for the paper appears to be "confirm Mickolajczyk but with bovine tubulin", and that's an under-sell.

– Several key figure panels not referred to in the text at all. E.g., I could not find a call out to Figure 1D and 1E, even though these figure panels are complex and difficult to understand. Please ensure that every figure panel is explained and called out individually. Figure 1D in particular is challenging for non-specialists.

– Many data points are missing error bars. See the bottom half of Figure 2. There should be error bars on the computational output as well, because stochastic simulations produce variable results. If the error is smaller than the displayed data point, please state so in the figure legend.

In addition, I recommend the following specific changes:

– The raw data kymographs are far too small (Figure 1B and Figure 3A). The quality of the raw IRM data looks very good, but it cannot be clearly assessed from such shrunken examples.

– When I compare Figure 1B (GMPCPP, low fluctuations) to Figure 3A (GTP, higher fluctuations), it's actually Figure 3A that appears to have smoother length vs. time plots. The issue is probably related to y-axis scaling, and I recommend consistent y-axes. In fact, it would be excellent to place GMCPP traces directly adjacent to GTP traces (perhaps in Figure 3), because that would allow for direct visualization of the core comparison being made in the manuscript. Figure 1-S1 does a nice job showing a side-by-side of pixel corrected vs. sub-pixel corrected.

– There is some inconsistency in the units used to describe bonds. E.g., Figure 2C uses molar affinities but Figure 2-S2 uses kT. I think in kT personally, but it's also fine if Figure 2-S2 shows affinity. Then the red arrow would point to 25 nM.

– Figure 2 S2 shows that the lateral bond strength parameter is not well specified, in the sense that the fitting residual is constant and minimal over at least a 4 kT range. Thus, the choice of lateral affinity (25 nM) is somewhat arbitrary, and the lateral affinity could be described as a "sloppy" parameter. Lateral bonds were also shown to be sloppy parameters that could be compressed out of a computational model of dynamic instability by Hsu et al., Biophys J 2020.

– Merge Figure 2 S4 and S5 --I understand the awkwardness there is that the other 3 show scaling of both k-on and K-D, while the best fit keeps the k-on of the "red" model and differs in K-D only. But right now I find it confusing, like there's a shadow parameter set that is not shown in some figures (main Figure 2, Figure 2 S3), but which is actually the most important parameter set. I encourage the authors to present all 4 parameter sets in a single, consistent format.

– In Figure 4, I recommend putting all of the model parameters in the figure itself. The parameters are currently found in the legend (4A), but the reader would find them more easily in the figure itself, and it would be clearer that the new parameters are NOT taken from previous figures, because of the new fitting to the GTP-growth data.

---

## [Author Response]

Essential revisions:1) Please provide more context and a broader motivation for the study in the Introduction making sure relevant literature is cited and shorten the summary of the results in the Introduction to avoid repetition.

We had internal discussions about the structure of our introduction before submitting the paper, so this comment did not exactly come as a surprise. We overhauled the introduction to provide more context about prior work and thinking, and to better motivate our experiments and analyses. As part of the re-write, we discussed the 1D model and explained why it is inadequate. We also shortened the summary of the results as suggested to limit repetition.

2) Model: Please explain if/why a 2D model is needed for microtubule growth (instead of only single filament model).

We addressed this point in our updated introduction. The 1D model oversimplifies the biochemistry because it assumes that there is only one kind of binding site on the microtubule end. But we know this is not true – the microtubule end can present multiple binding sites that differ in the number of contacts to neighboring tubulins. These different binding sites are expected to have substantially different affinities, but those different affinities cannot be obtained from the on- and off-rate constants obtained from a 1D model. We elaborate on this point in our response to one of the questions from reviewer 1, below.

3) Discussion: Please compare the model and the parameters used here to the relevant previous modelling studies, including a recent study of the authors (see reviewers' comments).

In responding to this comment, we focused on fits to the growth rates measured in GTP, because there are very few measurements of, and fits to, growth rates in GMPCPP.

We added a sentence in the Discussion to provide more commentary comparing our model to prior work in the area (from us and others), and to point out that we did not attempt to capture any mechanochemical features. Around line 422.

We expanded a second paragraph to compare the parameters obtained from fitting the growth rates in the presence of GTP to our prior fittings to other datasets. The fitted parameters are in the same general range that we and others have obtained, but more detailed comparison seems unwarranted because of differences in the measured growth rates between fitted sets and because of the different on-rate constants used (which are coupled to the longitudinal affinity). Around line 400.

4) Please make sure that all error bars are provided and clearly stated.

We have added error bars to Figures 2 B,E,F and Figure 4A. We also made sure that every figure legend states whether bars represent SD or SE.

5) Please make other editorial changes where appropriate to improve clarity of presentation, considering the suggestions made by the reviewers.

Done

Reviewer #1 (Recommendations for the authors):1. The Introduction currently provides a long summary of the results after a rather short introduction of previous work addressing the question of microtubule growth fluctuations. The topic could be introduced in a more comprehensive and a more conceptual manner and the summary of the results could in turn be shortened.

Thank you for the recommendation. As described in the overview, we overhauled the organization and information content of the introduction.

2. The Discussion could more clearly explain the difference between previous exclusively taper-shape dependent growth models and the model here which is a mixture of a taper shape and GTP hydrolysis model. Particularly why previous taper shape-dependent models that could explain even larger growth fluctuations now seem to fail for the reduced growth fluctuations of GTP microtubules as measured here.

We think there are a few issues to discuss here. There is the question of how fluctuations are determined from measurements. As we state in the text, our study used a more robust method to determine fluctuations than was available at the time of the earlier study *(Gardner et al., “Rapid microtubule assembly kinetics”, Cell 2011)* that we presume that the reviewer is referring to. That earlier study also used ‘concatenated’ growth episodes, which we showed leads to inflated estimates of fluctuations (Figure 1 —figure supplements 1-3). The earlier study introduced an on-rate penalty for lagging protofilaments – the way this was implemented generated ‘canyons’ (described in Castle and Odde, “Brownian dynamics of subunit addition-loss kinetics and thermodynamics in linear polymer self-assembly”, Biophys J 2013) that might contribute to elevated variations in microtubule growth rate for at least two reasons: first, because the presence of a canyon effectively reduced the number of protofilaments, and second because the presence of a canyon interrupted the pseudo-helical nature of the microtubule lattice (thereby introducing a barrier to elongation).

We added some text to be clearer about differences between models (around lines 311 and 364).

Reviewer #3 (Recommendations for the authors):The manuscript is rough and needs significant revisions:– The Introduction devotes too much text to summarizing the results (all of page 3 basically) and too little text to introducing the question and the state of the debate. It's really only the 2nd paragraph that introduces pertinent papers. The authors should reach further back than Gardner 2011, because Gardner 2011 was itself part of a series of studies whose intent was to measure the kinetics of polymerization. E.g. Kerssemakers et al., Nature 2006 and Schek et al., Curr Biol 2007 come to mind. Classic citations relating to association kinetics, like Northrup and Erickson PNAS 1992, would help contextualize the tubulin results in a broader biochemical context. As it stands, the motivation for the paper appears to be "confirm Mickolajczyk but with bovine tubulin", and that's an under-sell.

Thanks for these comments on the introduction. We took them to heart and substantially re-wrote the introduction to better introduce the questions and state of the debate.

– Several key figure panels not referred to in the text at all. E.g., I could not find a call out to Figure 1D and 1E, even though these figure panels are complex and difficult to understand. Please ensure that every figure panel is explained and called out individually. Figure 1D in particular is challenging for non-specialists.

We apologize for that. The manuscript has been updated to ensure that the figure panels in all main figures are referenced throughout the main text. (Line 162 for 1D with a reference to Castle 2019, line 167 for 1E )

– Many data points are missing error bars. See the bottom half of Figure 2. There should be error bars on the computational output as well, because stochastic simulations produce variable results. If the error is smaller than the displayed data point, please state so in the figure legend.

The computational figures have all been updated with error bars. In a few cases the bars are not visible because they are smaller than the symbols. We note this as suggested.

In addition, I recommend the following specific changes:– The raw data kymographs are far too small (Figure 1B and Figure 3A). The quality of the raw IRM data looks very good, but it cannot be clearly assessed from such shrunken examples.

The example kymographs shown in Figure 1B and 3A have been expanded as suggested.

– When I compare Figure 1B (GMPCPP, low fluctuations) to Figure 3A (GTP, higher fluctuations), it's actually Figure 3A that appears to have smoother length vs. time plots. The issue is probably related to y-axis scaling, and I recommend consistent y-axes. In fact, it would be excellent to place GMCPP traces directly adjacent to GTP traces (perhaps in Figure 3), because that would allow for direct visualization of the core comparison being made in the manuscript. Figure 1-S1 does a nice job showing a side-by-side of pixel corrected vs. sub-pixel corrected.

Direct comparison of growth traces is complicated in this case because of the different growth rates. To try to provide a more intuitive view of the magnitude of fluctuations, we added a new set of panels to Figure 3 that we hope provides a more intuitive view. This panel shows the fluctuations around the mean growth from the traces shown in Figure 1B and Figure 3A. We chose 1.5 μm GMPCPP and 7.5 μm GTP to because they had the closest growth rate rates of 2.6 nm/s and 5.04 nm/s respectively. As shown in figure 3 E, the faster the microtubule growth the higher the fluctuations. Here, we can see that a 2-fold change in velocity results in a >3fold change in fluctuations. (Add legend portion and figure callout in results Line number)

– There is some inconsistency in the units used to describe bonds. E.g., Figure 2C uses molar affinities but Figure 2-S2 uses kT. I think in kT personally, but it's also fine if Figure 2-S2 shows affinity. Then the red arrow would point to 25 nM.

To be consistent with the main figures, we updated Figure 2-S2 to use molar affinities on the x-axis. We agree with the reviewer that there are advantages to thinking in kT, but ultimately we prefer to use molar affinities because they are more directly relatable to experimental conditions.

– Figure 2 S2 shows that the lateral bond strength parameter is not well specified, in the sense that the fitting residual is constant and minimal over at least a 4 kT range. Thus, the choice of lateral affinity (25 nM) is somewhat arbitrary, and the lateral affinity could be described as a "sloppy" parameter. Lateral bonds were also shown to be sloppy parameters that could be compressed out of a computational model of dynamic instability by Hsu et al., Biophys J 2020.

We added a sentence to acknowledge this degeneracy. Line 242

– Merge Figure 2 S4 and S5 --I understand the awkwardness there is that the other 3 show scaling of both k-on and K-D, while the best fit keeps the k-on of the "red" model and differs in K-D only. But right now I find it confusing, like there's a shadow parameter set that is not shown in some figures (main Figure 2, Figure 2 S3), but which is actually the most important parameter set. I encourage the authors to present all 4 parameter sets in a single, consistent format.

Thanks for this suggestion. These supplemental figures have been updated, combined, and shown as a single figure with appropriate labeling.

– In Figure 4, I recommend putting all of the model parameters in the figure itself. The parameters are currently found in the legend (4A), but the reader would find them more easily in the figure itself, and it would be clearer that the new parameters are NOT taken from previous figures, because of the new fitting to the GTP-growth data.

We agree and it was an oversight to have not done this. We added the model parameters to the figure panel.